# Boron Neutron Capture Therapy (BNCT) Mediated by Maleimide-Functionalized *Closo*-Dodecaborate Albumin Conjugates (MID:BSA) for Oral Cancer: Biodistribution Studies and *In Vivo* BNCT in the Hamster Cheek Pouch Oral Cancer Model

**DOI:** 10.3390/life12071082

**Published:** 2022-07-20

**Authors:** Andrea Monti Hughes, Jessica A. Goldfinger, Mónica A. Palmieri, Paula Ramos, Iara S. Santa Cruz, Luciana De Leo, Marcela A. Garabalino, Silvia I. Thorp, Paula Curotto, Emiliano C. C. Pozzi, Kazuki Kawai, Shinichi Sato, María E. Itoiz, Verónica A. Trivillin, Juan S. Guidobono, Hiroyuki Nakamura, Amanda E. Schwint

**Affiliations:** 1Department of Radiobiology, National Atomic Energy Commission, Av. General Paz 1499, San Martin, Buenos Aires B1650KNA, Argentina; goldfingerjessica@gmail.com (J.A.G.); paula.s.ramos@outlook.com (P.R.); iarasofiasantacruz@gmail.com (I.S.S.C.); lulideleo@gmail.com (L.D.L.); garabalino@cnea.gov.ar (M.A.G.); trivilli@cnea.gov.ar (V.A.T.); or mandyschwint@gmail.com (A.E.S.); 2National Scientific and Technical Research Council (CONICET), Ciudad Autónoma de Buenos Aires C1425FQB, Argentina; 3Biodiversity and Experimental Biology Department, School of Exact and Natural Sciences, University of Buenos Aires, Av. Int. Güiraldes 2160, 4° piso, Pab. II, Ciudad Autónoma de Buenos Aires C1428EGA, Argentina; palmieri@bg.fcen.uba.ar; 4Department of Instrumentation and Control, National Atomic Energy Commission, Presbítero Juan González y Aragon, 15, Ezeiza, Buenos Aires B1802AYA, Argentina; thorp@cae.cnea.gov.ar; 5Department of Research and Production Reactors, National Atomic Energy Commission, Presbítero Juan González y Aragon, 15, Ezeiza, Buenos Aires B1802AYA, Argentina; paulacurotto@gmail.com (P.C.); epozzi@cnea.gov.ar (E.C.C.P.); 6School of Life Science and Technology, Tokyo Institute of Technology, Yokohama 226-8503, Japan; odckk831@gmail.com (K.K.); shinichi.sato.e3@tohoku.ac.jp (S.S.); hiro@res.titech.ac.jp (H.N.); 7Laboratory for Chemistry and Life Science, Institute of Innovative Research, Tokyo Institute of Technology, Yokohama 226-8503, Japan; 8Department of Oral Pathology, Faculty of Dentistry, University of Buenos Aires, M.T. de Alvear 2142, Ciudad Autónoma de Buenos Aires C1122AAH, Argentina; marielitoiz35@gmail.com; 9Buenos Aires Institute of Ecology, Genetics and Evolution (IEGEBA), CONICET, UBA, Intendente Güiraldes 2160, Ciudad Universitaria, Ciudad Autónoma de Buenos Aires C1428EGA, Argentina; jsguidobono@gmail.com

**Keywords:** MID:BSA, new boron compounds, BNCT, boron neutron capture therapy, oral cancer, hamster cheek pouch, mucositis

## Abstract

Background: BNCT (Boron Neutron Capture Therapy) is a tumor-selective particle radiotherapy that combines preferential boron accumulation in tumors and neutron irradiation. Although *p*-boronophenylalanine (BPA) has been clinically used, new boron compounds are needed for the advancement of BNCT. Based on previous studies in colon tumor-bearing mice, in this study, we evaluated MID:BSA (maleimide-functionalized *closo*-dodecaborate conjugated to bovine serum albumin) biodistribution and MID:BSA/BNCT therapeutic effect on tumors and associated radiotoxicity in the hamster cheek pouch oral cancer model. Methods: Biodistribution studies were performed at 30 mg B/kg and 15 mg B/kg (12 h and 19 h post-administration). MID:BSA/BNCT (15 mg B/kg, 19 h) was performed at three different absorbed doses to precancerous tissue. Results: MID:BSA 30 mg B/kg protocol induced high BSA toxicity. MID:BSA 15 mg B/kg injected at a slow rate was well-tolerated and reached therapeutically useful boron concentration values in the tumor and tumor/normal tissue ratios. The 19 h protocol exhibited significantly lower boron concentration values in blood. MID:BSA/BNCT exhibited a significant tumor response vs. the control group with no significant radiotoxicity. Conclusions: MID:BSA/BNCT would be therapeutically useful to treat oral cancer. BSA toxicity is a consideration when injecting a compound conjugated to BSA and depends on the animal model studied.

## 1. Introduction

Squamous cell carcinoma of the head and neck region is the seventh most common cause of cancer deaths worldwide [1]. One of the most frequent tumor sites is the oral cavity, the main risk factors being the consumption of alcohol, tobacco, and the oncogenic HPV (human papillomavirus) infection [1,2,3,4]. Despite the significant advances in diagnostic and treatment strategies, the five-year survival rate of oral cancer patients has not been improved, with the exception of some groups of HPV-positive oral carcinoma patients [3,5].

Despite modern advances in radiotherapy, there are head and neck cancer patients with high failure rates. An increase in the dose delivered to the tumor should improve tumor control, although it would also increase toxicity. Oral mucositis is a common toxicity of antineoplastic drugs and head and neck radiation in cancer patients that lacks an available effective treatment nowadays. This results from a multistage process, due to the damage of the therapy on the rapidly dividing cells of the epithelium and on the cells within the submucosa. This can cause erythema, edema, ulceration, necrosis, pain, dysphagia, and malnutrition. It affects patients’ quality of life and their treatment, because it leads to dose reductions, delays, and/or treatment interruptions [6,7,8]. Multimodality treatment could be an option, but in some cases, surgery or systemic therapies may be unsuitable due to different patient and tumor characteristics [3].

In this sense, there is a need for more selective therapies that deliver higher doses to the tumor, sparing the normal surrounding tissue during irradiation. BNCT (Boron Neutron Capture Therapy) is a targeted radiotherapy modality. It combines the administration of a drug containing ^10^B, a nonradioactive isotope of boron, which would accumulate preferentially in the tumor, with irradiation with a thermal or epithermal neutron beam. BNCT is a mixed-field radiation composed of ionizing radiation with different linear energy transfer (LET) characteristics. The tumor-specific boron dose component results from the high LET α (^4^He) particle and a recoiling lithium nucleus (^7^Li), which deposit their energy along a path of one cell diameter (approximately < 10 μm) and originate in the capture reaction of a thermal neutron by ^10^B. In addition to these high LET particles, there is a nonspecific background dose that similarly affects the tumor and normal surrounding tissues, e.g., [9,10]. As boron drugs accumulate preferentially in tumor cells and the high LET particles produced by the capture reaction have a short range, most of the radiation effect would occur in these cells, making BNCT a biologically rather than geometrically targeted therapy [11,12].

Worldwide, BNCT clinical studies have been reported principally for brain tumors, head and neck cancer, and melanomas, e.g., [13,14,15,16,17,18,19]. To date, clinical results have shown positive therapeutic effects with room for improvement. In this sense, the development of new, more tumor-selective, non-toxic, and effective boron compounds is a current need, which would increase BNCT therapeutic efficacy and would reduce the time of irradiation, consequently reducing the background dose that contributes to unwanted side effects in normal tissue [7].

Animal models are essential for studying BNCT efficacy and toxicities. Albeit with limitations, translational studies serve as a bridge between basic science and clinical studies [11]. The hamster cheek pouch has been widely described and accepted as one of the most ideal animal models to study oral cancer, therapeutic alternatives, and mucositis [20,21,22]. Unlike models of implanted tumor cells in normal tissue, it provides a tumor model surrounded by a precancerous dose-limiting tissue, from which additional tumors develop, mimicking the events involved in the development of premalignant and malignant human oral lesions [23]. The hamster cheek pouch is similar to the human oral mucosa (in histological, histochemical, and ultrastructural terms), does not develop spontaneous tumors, and is easily accessible *in situ*. The protocol used to induce these tumors consists of the topical application of subthreshold doses of the complete carcinogen DMBA (7,12-dimethylbenz[a]anthracene), simulating the action of alcohol and smoking in humans [24,25]. In our group, we employed the hamster cheek pouch model to study the therapeutic effect of BNCT on tumors and the mucositis induced in the precancerous tissue surrounding tumors, employing different boron compounds approved for their use in humans and novel boron carriers, e.g., [10,26,27,28,29].

Recently, nano carrier-based boron delivery systems have been developed to improve the efficacy of BNCT, e.g., [30]. The focus is on serum albumin as a nano biocarrier. Albumin is a major plasma protein constituent, distributed between the blood circulation, the lymphatic system, and the extracellular as well as intracellular compartments. It is one of the most important drug carriers, especially for the treatment and diagnosis of malignant, inflammatory, metabolic, and viral diseases, as it accumulates due to the EPR (enhanced permeability and retention) effect [30,31,32]. Moreover, the uptake of extracellular proteins is markedly activated in rapidly growing tumor cells, albumin being the major nutritional source for the tumor and, in this way, the major site of serum albumin catabolism [33].

In the nano biocarrier approach using serum albumin, maleimide-functionalized *closo*-dodecaborate (MID) was developed and introduced into bovine serum albumin (BSA). MID was found to conjugate to lysine residues as well as the free SH of cysteine residues in BSA under physiological conditions. The highly boronated MID:BSA at 30 mg B/kg showed high and selective accumulation in the tumor 12 h post-administration. When it was used as a boron carrier for BNCT, a significant therapeutic effect was observed in colon tumor-bearing mice [34]. Furthermore, the MID–human serum albumin conjugates showed BNCT therapeutic efficacy equivalent to that of BPA against the F98 rat glioma model [35]. The aim of the present study was to assess MID:BSA biodistribution in the hamster cheek pouch oral cancer model and to perform dose-escalation studies to evaluate the therapeutic and radiotoxic effects of BNCT mediated by MID:BSA in this experimental oral cancer model.

## 2. Materials and Methods 

For all our experiments, six- to eight-week-old Syrian hamsters were exposed to the optimized classical carcinogenesis protocol [36]: the topical application of DMBA in mineral oil (0.5%) in the right cheek pouch twice a week for 12 weeks, with 2 interruptions (4^th^ and 5^th^ application) completed at the end of the protocol. In previous studies [36], this optimized protocol reduced the mucositis induced by the cancerization protocol. The animals were then assigned to different experimental groups for boron biodistribution studies, BNCT radiobiological studies, and to a control group (cancerized sham-irradiated, i.e., matched manipulation, no treatment).

Experiments were carried out in accordance with the guidelines laid down by the National Institute of Health in the USA regarding the care and use of animals for experimental procedures. Experiments were designed based on the 3Rs (replacement, refinement, and reduction principles), evaluated, and approved by the Argentine National Atomic Energy Commission Animal Care and Use Committee (CICUAL-CNEA, n° 11/2018). Tap water and a standard diet (Cooperación, Argentina) were supplied *ad libitum*. The room temperature was about 24 °C with a 12/12 h light/dark cycle. Cage changing was performed three times per week. The treated pouch was periodically everted under intraperitoneal (i.p.) anesthesia (ketamine (140 mg⁄kg) and xylazine (21 mg/kg)) to examine tumor development and mucositis during and after cancerization, pre and post-BNCT. The intravenous (i.v.) injection of MID:BSA was performed in the jugular vein under i.p. anesthesia (ketamine (70 mg/kg) and xylazine (10.5 mg/kg)). Analgesia was performed with Tramadol (6 mg/kg/day, i.p.) when injecting MID:BSA i.v. (the day before and 48 h after surgery) or if the animals exhibited severe mucositis (until resolution).

### 2.1. MID:BSA Preparation and Biodistribution Studies

MID was conjugated to BSA (Sigma-Aldrich-Merck, Darmstadt, Germany) in phosphate-buffered saline (PBS) by stirring the solution during 12 h at 37 °C. The final solution was yellowish and clear. We explored two MID:BSA concentration protocols: 30 mg B/kg (21.1 mg MID: 307 mg BSA) and 15 mg B/kg (10.5 mg MID:154 mg BSA), injected i.v. into the surgically exposed jugular vein under light anesthesia. MID:BSA intravenous injection was performed at a very slow rate to avoid acute toxicity caused by BSA, i.e., 0.05–0.1 mL every 30/60 s. As described in the Results section, pilot studies with MID:BSA at 30 mg B/kg showed that this concentration of BSA was toxic for the hamsters. Thereafter, we continued our studies, employing the 15 mg B/kg protocol, at two different times after injection: 12 h (“12 h-protocol”, 4 animals) and 19 h (“19 h-protocol”, 8 animals).

For both protocols, 12 h and 19 h, samples of blood, tumor, precancerous and normal pouch tissue, and, as references, spleen, liver, and kidney were processed for gross boron concentration measurements by ICP-OES as in Garabalino et al. [37]. We calculated, for each tissue, the mean boron concentration values (ppm ± SD) and the tumor/precancerous tissue, tumor/normal tissue, and tumor/blood boron concentration ratios for each protocol.

### 2.2. Preliminary MID:BSA Immunostaining Studies

Two of the animals injected with MID:BSA (15 mg B/kg, 19 h-protocol) for biodistribution studies were also used to analyze MID:BSA microlocalization by immunostaining. The cancerized pouch-bearing tumors surrounded by precancerous tissue and the normal contralateral pouch (non-cancerized) were removed and fixed in 10% buffered formalin and then paraffin-embedded. The tissue sections of the paraffin-fixed tissue sample (8 µm thickness) were deparaffinized by washing these sections in lemosol for 5 min, three times, and in EtOH for 5 min, three times. Then, these sections were washed sequentially with 90% EtOH, 80% EtOH, 70% EtOH, and PBS. Sections were treated with 5% BSA/PBS and were washed with TTBS (Tween-Tris-buffered saline: Tris-buffered saline with 0.05% Tween 20) three times. The sections were treated with an anti-MID rabbit antibody (500 times diluted) for 2 h followed by a secondary antibody (anti-rabbit IgG biotinylated antibody (Cell Signaling Technology, Danvers, MA, USA) or goat anti-rabbit IgG-Texas Red (Abcam, Boston, MA, USA; 500 times diluted)) for 45 min. After washing by TTBS three times, the sample was enclosed with VECTASHIELD mounting medium (VECTOR Laboratories, Newark, CA, USA). Fluorescence images were observed by fluorescence microscopy (Olympus). Adjacent sections were stained with hematoxylin-eosin (HE) according to the same protocol to examine MID:BSA microdistribution in the tumor, precancerous tissue (Pr), normal tissue (N), connective tissue (C), and muscle (M).

### 2.3. In Vivo BNCT Studies

Based on our biodistribution studies, we chose the MID:BSA 15 mg B/kg, 19 h protocol to perform MID/BSA-BNCT dose-escalation studies in the hamster cheek pouch oral cancer model. We evaluated 4 groups of cancerized hamsters: (CONTROL) cancerized, no irradiation (n = 8 animals); BNCT mediated by MID:BSA (15 mg B/kg, 19 h) at: (GROUP A) 4.8 Gy absorbed dose to precancerous tissue (n = 3 animals); (GROUP B) 6.0 Gy absorbed dose to precancerous tissue (n = 3 animals); and (GROUP C) 7.5 Gy absorbed dose to precancerous tissue (n = 4 animals).

Irradiations were performed at the RA-3 nuclear reactor thermal facility (Ezeiza, Buenos Aires, Argentina). To protect the body of the animals from thermal neutrons, we employed a ^6^Li carbonate shielding, everting the cheek pouch-bearing tumors out of the enclosure onto a protruding shelf for exposure. Total absorbed dose was prescribed to the dose-limiting tissue, i.e., the precancerous tissue around tumors, based on toxicity data obtained in previous BNCT studies with other boron compounds, e.g., [7,10,27,29]. Irradiation conditions are reported in Table 1. Dose calculations were performed considering the boron concentration value in precancerous tissue reported in Table 2 for the MID:BSA 15 mg B/kg, 19 h-protocol, i.e., 24.3 ± 11.3 ppm.

During follow-up after BNCT, we assessed weekly the clinical signs and body weight of the animals. Oral mucositis in each group was evaluated semi-quantitatively according to an adaptation of oral mucositis scales in hamsters and humans [7,38,39]: Grade 0: healthy appearance, no erosion or vasodilation; Grade 1 (slight): erythema and/or edema and/or vasodilation, no evidence of mucosal erosion; Grade 2 (slight): severe erythema and/or edema, vasodilation and/or superficial erosion; Grade 3 (moderate): severe erythema and/or edema, vasodilation, and the formation of ulcers < 2 mm in diameter; Grade 4 (severe): severe erythema and/or edema, vasodilation, and the formation of ulcers ≥ 2 mm and <4 mm in diameter, and/or necrosis areas < 4 mm in diameter; Grade 5 (severe): the formation of ulcers ≥ 4 mm in diameter or multiple ulcers ≥ 2 mm in diameter, and/or necrosis areas ≥ 4 mm in diameter. Grading was based on the most severe macroscopic feature.

We evaluated the therapeutic effect of BNCT on those tumors that were present at the time of irradiation (T0) during 28 days. We considered those exophytic tumors that reached a volume of ≥1 mm^3^ and ≥0.7 mm in height [36]. Tumor volume was determined by external caliper measurement of the three largest orthogonal diameters (d) and calculated as d1 × d2 × d3, e.g., [40]. Tumors were measured at 7 days, 14 days, 21 days, and 28 days after BNCT and we assessed the % of tumors with complete response (CR: the disappearance of the tumor on visual inspection); the % of tumors with partial response (PR: reduction in pre-treatment tumor volume); the % of tumors with no response (NR); and the % of tumors with an overall response (OR) = partial response (PR) + complete response (CR). We also calculated the percentage of partially responding tumors that reduced their volume by more than/equal to 50% (R ≥ 50%) from the T0 value for each time-point.

### 2.4. Statistical Analysis

Statistical analyses were carried out using the R statistical program [41]. The difference in gross boron content between 12 h and 19 h for each tissue was evaluated by Student’s *t*-test. Tumor/precancerous tissue, tumor/normal tissue, and tumor/blood for 12 h vs. 19 h were evaluated with a generalized least squares model using the identity variance function structure [42]. OR and R > 50% between the CONTROL group and BNCT groups for all time-points were analyzed using a mixed log-binomial test [43]. Pairwise comparisons for proportions of OR and R > 50% between the MID:BSA/BNCT groups were evaluated using the Bonferroni method for adjusting *p*-values. *p*-values of less than 0.05 were considered statistically significant. *p*-values between 0.1 and 0.05 were regarded as a statistical trend.

## 3. Results

All MID:BSA protocols induced an increase in the hamster pulse rate, jugular vein swelling at the site of injection, and an increase in the breathing frequency. In the case of the 15 mg B/kg protocol, these adverse effects could be reverted by stopping the injection or reducing the injection rate to 0.05–0.1 mL every 30/60 s. However, the hamsters injected with 30 mg B/kg MID:BSA and BSA only (307 mg, i.v.) died during the injection. They exhibited high BSA toxicity, i.e., internal bleeding and cardiorespiratory arrest.

Table 2 shows the mean boron concentration values for each tissue at 12 h and 19 h after MID:BSA 15 mg B/kg i.v. injection. The mean boron concentration value in the tumor at 12 h was therapeutically useful (32.4 ± 9.0 ppm), but the mean blood boron concentration (41.9 ± 6.8 ppm) was higher than in the tumor. To study boron clearance from blood, we decided to evaluate tissue samples at 19 h after injection. At 19 h post-injection, the mean boron concentration in the tumor (31.7 ± 11.1 ppm) did not change significantly vs. the 12 h protocol, while blood boron concentration (23.2 ± 9.0 ppm) fell significantly (*p* = 0.00474) vs. the 12 h protocol. For both the 12 h and 19 h protocols, the tumor/normal tissue ratio revealed a preferential boron uptake by the tumor vs. normal tissue. In the case of the tumor/precancerous tissue ratio, the ratio was lower at 19 h post injection than at 12 h. This difference is marginally significant (*p* = 0.0925). However, although the ratio was lower, we still observed preferential (albeit not statistically significant) boron uptake in the tumor. The fact that the preferential uptake in tumor vs. precancerous tissue did not reach statistical significance would be due to the spread in values.

**Table 2 life-12-01082-t002:** Boron concentration in ppm (mean ± SD) for each tissue and tumor/blood, tumor/precancerous, and tumor/normal tissue boron concentration ratios (calculated as in [44]) for MID:BSA 15 mg B/kg at 12 h and 19 h post-injection. One to two samples were measured per tissue. Each animal bears 2 to 8 tumors, approximately. Exceptionally, in the case of the 19 h protocol, one of the animals did not exhibit tumors.

	MID:BSA 15 mg B/kg 12 h-Protocol		MID:BSA 15 mg B/kg 19 h-Protocol	
**Tissue**	**n**	**ppm ± SD**	**n**	**ppm ± SD**
Tumor	15 tumors	32.4 ± 9.0	27 tumors	31.7 ± 11.1
Precancerous tissue	4 animals	15.7 ± 3.0	8 animals	24.3 ± 11.3
Normal	4 animals	5.5 ± 1.3	8 animals	6.8 ± 1.5
Liver	4 animals	27.4 ± 5.3	8 animals	30.4 ± 12.7
Kidney	4 animals	25.2 ± 3.9	8 animals	27.5 ± 7.6
Spleen	4 animals	14.0 ± 3.7	4 animals	17.7 ± 3.8
Blood	4 animals	41.9 ± 6.8	8 animals	23.2 ± 9.0
**Tumor/precancerous tissue**	**2.1 ± 0.6**		**1.3 ± 0.6**	
**Tumor/normal tissue**	**6.1 ± 0.9**		**4.7 ± 1.4**	
**Tumor/blood**	**0.8 ± 0.1**		**1.6 ± 1.1**	

Immunostaining studies allowed us to evaluate qualitatively the distribution of MID:BSA in tumors, precancerous tissue, and normal tissue. Figure 1 shows two different tumors with different MID:BSA microdistributions: in tumor 1, the parenchyma had lower MID:BSA accumulation vs. the stroma. Meanwhile, in tumor 2, no differences were observed between the parenchyma and stroma.

Figure 2 shows MID:BSA microdistribution in a cancerized pouch and in a normal pouch. In both samples, the connective tissue exhibited higher MID:BSA uptake versus the muscle fascicules. When comparing the precancerous epithelium vs. normal epithelium, we observed a heterogeneous MID:BSA distribution in the precancerous epithelium, whereas the normal epithelium exhibited no MID:BSA uptake. The reduced number of samples evaluated allows us to describe these findings as contributory but preliminary.

After BNCT mediated by MID:BSA/BNCT 15 mg B/kg (19 h protocol), we observed that only at 7.5 Gy, one of the three irradiated animals exhibited Grade 4 mucositis, albeit without necrosis. None of the remaining animals developed severe mucositis. In addition, BNCT did not exacerbate the pre-BNCT mucositis that the animals exhibited due to the cancerization process [36].

As for the tumor response to BNCT, Table 3 shows the percentage of tumors that exhibited OR for the control, A (4.8 Gy), B (6.0 Gy), and C (7.5 Gy) groups vs. time. For all time points, the BNCT groups exhibited a significantly higher %OR vs. the control group (*p* < 0.001). As for the %CR of BNCT groups vs. control group, for all time-points, we observed statistically significant differences for Groups A and C (*p* = 0.03 and *p* = 0.006, respectively) and a statistical trend for Group B (*p* = 0.07).

Regarding the MID:BSA/BNCT groups, we observed that at 7 days post-BNCT, Group B exhibited 100% of tumors with OR. This value was significantly higher than for Group A (54%, *p* = 0.002) but it did not differ significantly from Group C (78%). As for R ≥ 50% (Table 4), at 7 days after BNCT, Group B exhibited the highest value vs. Groups A and C, although this difference was not significant. At 14 and 21 days after BNCT, the %OR and R ≥ 50% for groups B and C were similar and higher (although not statistically significantly) than Group A (Table 3 and Table 4). For the parameter R ≥ 50% at 14 days after BNCT, Group C exhibited a similar R ≥ 50% value compared to Group B. However, a statistical trend (*p* = 0.068) was observed comparing Group C vs. Group A. At 28 days after BNCT, Group B exhibited a higher %OR versus Groups A and C, although this difference was not statistically significant (Table 3, 82% vs. 59% y 67%, respectively). For R ≥ 50% (Table 4), Group C exhibited a higher value than Groups A and B (92% vs. 55% y 64%, respectively), although this difference was not statistically significant.

## 4. Discussion

For the first time in the hamster cheek pouch oral cancer model, the boron compound MID conjugated to BSA was evaluated in terms of its biodistribution and therapeutic BNCT effect on tumors and radiotoxicity. For that aim, we first evaluated the 30 mg B /kg, 12 h protocol studied in a colon cancer model in mice [34]. In our studies, the hamsters did not tolerate the 30 mg B/kg protocol due to BSA toxicity, suggesting that BSA toxicity would depend on the animal model. The US Food and Drug Administration established recommendations on human albumin dosage and administration [45]. Albumin solution concentration, dosage, and infusion rate should be adjusted to the patient’s individual requirements and, during administration, hemodynamic performance should be monitored regularly including arterial blood pressure, pulse rate, and central venous pressure, among other parameters. In our experiment, as we injected MID:BSA intravenously in the surgically exposed jugular vein, we were able to macroscopically monitor jugular vein swelling, pulse rate, and breathing frequency. These parameters allowed us to define, for the hamster, the optimal BSA concentration and injection rate that was reported and used in this study.

Having demonstrated that the animals tolerated the MID:BSA 15 mg B/kg protocol, we performed ensuing studies with this administration protocol. Biodistribution studies were carried out at 12 h and 19 h after MID:BSA administration. Based on previous radiobiological BNCT studies in the hamster cheek pouch oral cancer model employing the boron compounds BPA and GB-10, we defined guidelines to establish the potential therapeutic value of the administration protocols assessed: no manifest toxicity; absolute boron concentration in tumor ≥ 20 ppm; boron concentration ratio tumor/normal tissue ≥ 1; and boron concentration ratio tumor/blood ≥ 1 [44]. For the MID:BSA 15 mg B/kg protocol at 12 h after injection, we found a therapeutically useful boron concentration value in the tumor and tumor/normal tissue ratio. However, the boron concentration in blood was higher than in the tumor, resulting in a tumor/blood concentration ratio lower than 1. Instead, for the 19 h protocol, the boron concentration value in blood was reduced with no significant reduction in the absolute boron concentration in the tumor, and the tumor/normal tissue boron concentration ratio remained high. These results imply that it would be possible to maximize the delivery of boron to the tumor and minimize boron concentration in normal tissue and blood, thus reducing the irradiation times and, consequently, the nonspecific background dose that affects tumor and normal tissue similarly [11].

We also evaluated the boron uptake in the precancerous tissue surrounding tumors. In our study, we observed higher boron absolute concentration values in the 19 h protocol vs. the 12 h protocol, although this difference was not statistically significant. For both protocols, the tumor/precancerous tissue boron concentration ratio was higher than 1. This is an important result, as BNCT can induce mucositis that could affect animal welfare and limit the dose delivered to the tumor, e.g., [7]. However, we observed a greater spread in boron concentration values expressed in terms of standard deviation/mean percentage in precancerous tissue for the 19 h protocol (46.5%) vs. the 12 h protocol (19.1%), suggesting an increase in the degree of heterogeneity in boron accumulation for the 19 h protocol.

Biodistribution studies also showed that precancerous tissue boron concentration values are, overall, higher than normal pouch tissue values. This result would make this protocol potentially useful to achieve a therapeutic effect in precancerous tissue in terms of tumor development inhibition without significant damage to normal pouch tissue, e.g., [46,47].

It is known that the compounds conjugated to albumin accumulate in tumors due to the EPR (enhanced permeability and retention) effect, which occurs due to the combination of leaky capillaries with the absence or defect of the lymphatic drainage system. The hamster cheek pouch oral cancer model mimics the spontaneous process of malignant transformation. In the same treated pouch, different stages of evolution of the carcinogenesis process can coexist, i.e., epithelium with no unusual microscopic features (NUMF), hyperplasia, dysplasia, and exophytic and endophytic tumors [46]. In this animal model, Aromando et al. [48] demonstrated that the angiogenic switch precedes tumor development and, during carcinogenesis, the blood vessels underlying dysplastic lesions exhibit abnormal morphology. This altered vascular morphology contributes to the EPR effect, not only in tumors but also in the connective tissue underlying precancerous tissue.

In addition to the EPR effect, the gp60 receptor located in tumor blood vessel endothelium, which transports albumin through transcytosis into the tumor interstitium, contributes to MID:BSA accumulation in tumors. Furthermore, SPARC proteins (Secreted Protein, Acidic, and Rich in Cysteine) that bind to the albumin-bound drugs in the tumor extracellular matrix contribute to a homogeneous tumor tissue distribution [31]. Particularly, SPARC proteins were detected in oral squamous cell carcinoma [49]. Aquino et al. [49] highlighted that the main activity of SPARC proteins occurs in the tumor microenvironment. Their aberrant expression was observed in stromal cells surrounding the tumor. Finally, Chen et al. [50] demonstrated that SPARC proteins were up-regulated in DMBA-cancerized hamster cheek pouches.

Absolute boron content and the microdistribution of the boron compound in the tumor, precancerous tissue, and healthy tissues are central to the efficacy of BNCT. Based on the described biodistribution results, we performed a preliminary boron microdistribution study in tumor, precancerous, and normal tissue. Gross boron measurements do not allow us to identify boron concentration in the epithelium, separately from stroma/connective tissue and muscle. Although more boron microdistribution studies are needed, we observed in this preliminary study that MID:BSA accumulated in tumor stroma and in some areas of tumor parenchyma. In the precancerous tissue surrounding the tumors, MID:BSA preferentially accumulated in the connective tissue, although some regions of the epithelium were also stained. Conversely, no accumulation was seen in the normal, non-cancerized epithelium. While these data are preliminary and show variability in MID:BSA microdistribution in tumors and in precancerous tissue, they do suggest preferential accumulation in cancerized epithelium vs. normal epithelium.

Our results showed MID:BSA microdistribution variability and a large spread in the gross boron absolute concentration values. Although further studies are needed, it is known that there are several factors that could condition MID:BSA distribution. For example, lesions with different degrees of malignancy would have different MID:BSA uptakes, as the uptake of extracellular proteins such as serum albumin is markedly activated in rapidly growing tumor cells [33,34]. Moreover, Park et al. [51] explained that the heterogeneity of the EPR effect depends on the tumor microenvironment, i.e., the characteristics of the extracellular matrix, the distribution of blood flow, and blood vessel permeability.

Based on the biodistribution studies described above, we studied BNCT mediated by MID:BSA 15 mg B/kg, 19 h protocol. This protocol was therapeutically useful to treat tumors in the hamster cheek pouch oral cancer model with no significant toxicity. At 7 days, we observed that the MID:BSA/BNCT 6.0 Gy (Group B) exhibited 100% OR and the highest R ≥ 50% vs. the 4.8 Gy (A) and 7.5 Gy (C) groups. However, at this time point, only the percentage of tumors with OR 6.0 Gy vs. 4.8 Gy reached a statistically significant difference. At 14, 21, and 28 days after BNCT, we observed that Group B exhibited the highest %OR, although it was not significantly different from the other groups. Our results suggest that MID:BSA/BNCT exhibits a therapeutic threshold, as higher doses do not correlate with higher tumor control. Park et al. [51] explained that the therapeutic effect with compounds that accumulate in the tumor by the EPR effect is not always satisfactory. The heterogeneity of the EPR effect depends on the tumor microenvironment, which may vary within the same tumor (at different tumor stages and over time) and also among individual tumors. Depending on the characteristics of the tumor microenvironment, the penetration of nanomedicines from the vessels deep into the tumor interstitium could be impaired [52]. In our hamster model, at the time of irradiation, tumors with different characteristics are present in the same treated pouch, possibly accounting for varying degrees of tumor response.

Regarding BNCT-induced mucositis in precancerous tissue, we did observe an increase in mucositis severity with the dose delivered to precancerous tissue. As we explained before, MID:BSA accumulates in precancerous tissue by different mechanisms. In addition, Fang et al. [53] mentioned that inflammatory tissues also exhibited the EPR effect. In our studies, all animals exhibited mucositis pre-BNCT as a result of cancerization. However, although MID:BSA accumulates in precancerous tissue, only high doses (7.5 Gy) induced severe mucositis (with no necrosis) in one of the animals. Within the context of cancer treatment, mucositis develops when oncological therapies affect the dividing cells of the epithelium and the cells and tissues within the submucosa. Particularly in BNCT, the boron compound BPA accumulates in these cells due to their increased metabolism, causing BNCT mediated by BPA to induce severe mucositis vs. other protocols, e.g., [7,47]. In that sense, low boron concentration in non-tumor cells of the precancerous tissue surrounding tumors would be an asset in terms of reducing radiotoxicity. However, we also demonstrated that BNCT mediated by BPA inhibits the long-term tumor development in precancerous tissue that mimics the development of second primary tumors in field-cancerized tissue in humans. This result, associated with no severe mucositis in dose-limiting precancerous tissue, would be due to the increase in the capability of those potentially malignant and tumor cells present in the precancerous tissue to incorporate more BPA, due to their high metabolism, than the rest of the non-tumor cells in this tissue [47]. In this study, we observed a heterogeneous distribution of MID:BSA in precancerous tissue and no accumulation in the normal tissue. Based on the tumor uptake mechanisms of MID:BSA explained above, we could suggest that MID:BSA is incorporated preferentially by those potentially malignant and tumor cells that are present in the precancerous tissue vs. the remaining non-tumor cells in the precancerous and normal tissue. This would be an important result, indicating that MID:BSA would be safe in terms of radiotoxicity and would make this boron agent potentially useful to inhibit the development of new tumors in precancerous tissue. More microdistribution studies are needed to confirm this hypothesis.

We also compared our results with other BNCT studies performed previously in our group, in tumor-bearing hamsters, employing boron compounds that accumulate by a similar mechanism to MID:BSA, the EPR effect, such as MAC-TAC liposomes [29]. Comparing MID:BSA/BNCT at 4.8 Gy vs. MAC-TAC liposomes at a 5.0 Gy absorbed dose to precancerous tissue, the percentage of tumor overall responses at 28 days were similar, i.e., 59% vs. 64%, respectively. In both cases, no mucositis was observed in precancerous tissue.

MID:BSA/BNCT proved to be therapeutically useful for the treatment of oral cancer. However, there is room for improvement. Dhaliwal and Zheng [54] suggest that, to improve the EPR effect to facilitate nanomedicine delivery, EPR-adaptive strategies should be employed. These strategies consist of modifying tumor accessibility by combining nanomedicine delivery with chemical or physical techniques to increase agent delivery and therapeutic effect. In this sense, for future studies, rather than increasing the dose to enhance tumor control, it is necessary to combine MID:BSA/BNCT with other strategies and boron compounds with different uptake mechanisms. For example, in previous studies by our group, e.g., [10] we demonstrated that the combination of the boron compound BPA (boronophenylalanine) + GB-10 improved absolute and relative boron concentration and favored homogenous boron targeting in tumors, exerting a significantly higher therapeutic BNCT effect on tumors vs. BNCT mediated by BPA or GB-10 alone, with no severe mucositis in precancerous tissue. BPA accumulates preferentially in tumor parenchyma. Although GB-10 is not incorporated selectively into oral tumors, BNCT mediated by GB-10 exerts a selective effect on the tumor by acting on the more radiosensitive aberrant tumor vasculature [10]. The combination of different compounds with different uptake mechanisms and properties would deliver large amounts of boron to the tumor and significantly improve boron delivery homogeneity in the tumor, i.e., boron would target all tumor cell subpopulations. The combined administration of boron compounds would conceivably allow higher-absorbed doses to be delivered more homogenously to the tumor, prescribing lower doses to the precancerous dose-limiting tissue, increasing BNCT therapeutic efficacy on tumors and reducing the BNCT-induced mucositis in precancerous tissue.

Another possible strategy to explore, which could enhance MID:BSA/BNCT therapeutic effect, is to perform a second application of BNCT. This strategy is considered an EPR-adaptive strategy in radiotherapy [54] and our group proposed and demonstrated the therapeutic advantage of Sequential BNCT (with a 24/48 h interval between irradiations) in the oral cancer model [40]. Interstitial fluid pressure (IFP) in the tumor is elevated due to the upregulated angiogenesis, abnormal blood vessels, the lack of functional lymphatic vessels, tumor cell proliferation, and the characteristics of the tumor microenvironment. Elevated IFP would impair the distribution of boron compounds in the tumor. The first application would reduce IFP due to vascular regression, the death of tumor cells that compress microvasculature, and decrease resistance to blood flow, allowing more convective movement towards the tumor core [40,54] and favoring tumor boron targeting in the second application.

In previous studies by our group, we explored Double BNCT, i.e., two full-dose BNCT treatments with an interval of 2–6 weeks. In this case, this strategy would also allow the retargeting of those tumor cell populations that were refractory to the first BNCT treatment, increasing the therapeutic effect of BNCT on tumors at later times after irradiation without exacerbating dose-limiting mucositis. Our group demonstrated that a double BNCT did not exacerbate mucositis in precancerous tissue, increased the BNCT therapeutic effect on tumors, and inhibited tumor development in precancerous tissue, e.g., [29,55]. MID:BSA/BNCT at 6.0 Gy would exhibit an advantage versus the other dose levels examined herein, in terms of comparable (or higher) and useful therapeutic effects on tumors without associated severe mucositis in precancerous tissue. In this sense, a second MID:BSA/BNCT at 6.0 Gy could conceivably be performed at 7 days after BNCT.

## 5. Conclusions

The results of this study evidenced that BNCT mediated by MID:BSA would be therapeutically useful to treat oral cancer with no significant toxicity in the dose-limiting tissue surrounding tumors, the precancerous tissue. The study of new boron compounds and strategies is a current need to improve the efficacy of BNCT for different pathologies, particularly for strategies that depend on the EPR effect. When studying BSA conjugated to a boron compound, BSA toxicity is a consideration and depends on the animal model evaluated.

## Figures and Tables

**Figure 1 life-12-01082-f001:**
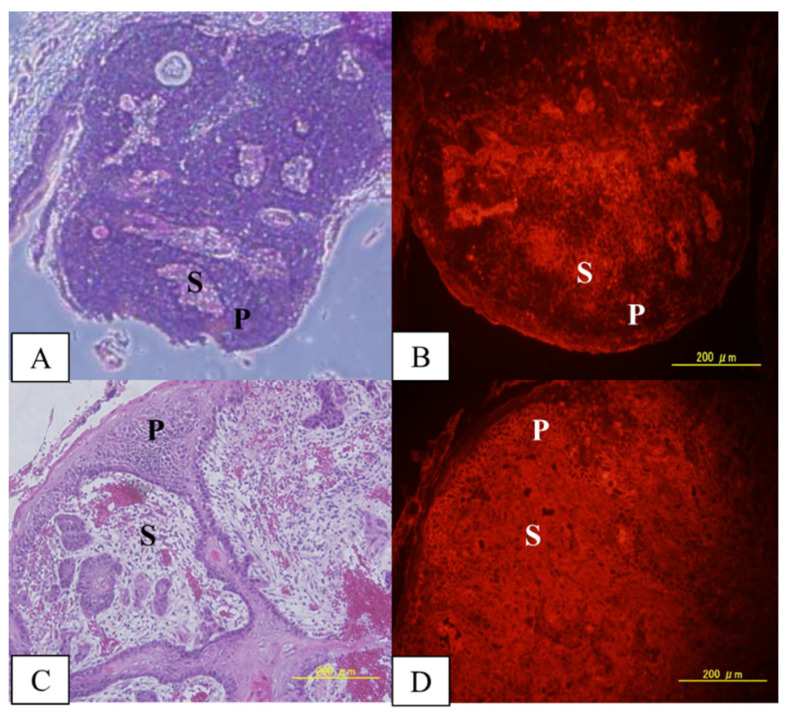
MID:BSA immunostaining in 2 different tumors, showing different microdistributions: (**A**,**B**) H&E of tumor 1 and its corresponding MID:BSA immunostaining; (**C**,**D**) H&E of tumor 2 and its corresponding MID:BSA immunostaining. A brighter image is indicative of a higher boron concentration. P: parenchyma; S: stroma.

**Figure 2 life-12-01082-f002:**
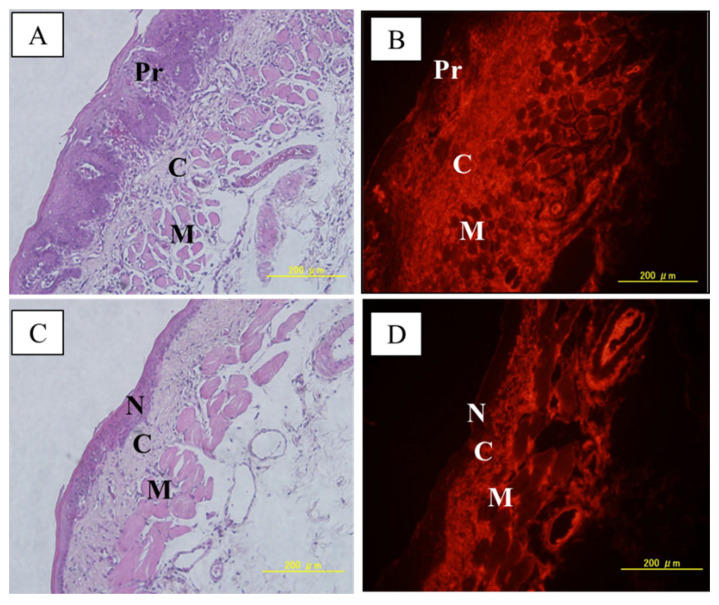
(**A**,**B**): H&E of a precancerous tissue with its corresponding immunostaining. (**C**,**D**) H&E of normal tissue with its corresponding immunostaining. A brighter image is indicative of a higher boron concentration. Pr: precancerous tissue; C: connective tissue; M: muscle; N: normal tissue.

**Table 1 life-12-01082-t001:** Irradiation conditions. The dose in each BNCT protocol was prescribed to precancerous tissue, the dose-limiting tissue. * Components of background dose; ** Total background dose.

Dose Components	Fast Neutrons *	Gamma Photons *	Induced Protons *	Total Dose Without Boron Irradiation Component **	Boron Irradiation Component	Boron Dose/ppm Boron Concentration	Total Absorbed Dose
**MID:BSA/BNCT 15 mg/kg 19 h post-injection-4.8 Gy absorbed dose to precancerous tissue**							
Tumor	-	0.44 ± 0.04 Gy	0.60 ± 0.13 Gy	1.05 ± 0.14 Gy	4.9 ± 2.0 Gy	0.16 ± 0.08 Gy	**6.0 ± 2.0 Gy**
Precancerous tissue	-	0.44 ± 0.04 Gy	0.60 ± 0.13 Gy	1.05 ± 0.14 Gy	3.8 ± 1.9 Gy	0.16 ± 0.11 Gy	**4.8 ± 1.9 Gy**
Normal tissue	-	0.44 ± 0.04 Gy	0.60 ± 0.13 Gy	1.05 ± 0.14 Gy	1.1 ± 0.3 Gy	0.16 ± 0.06 Gy	**2.1 ± 0.4 Gy**
**MID:BSA/BNCT 15 mg/kg 19 h post-injection-6.0 Gy absorbed dose to precancerous tissue**							
Tumor	-	0.55 ± 0.06 Gy	0.74 ± 0.16 Gy	1.30 ± 0.17 Gy	6.1 ± 2.5 Gy	0.19 ± 0.10 Gy	**7.4 ± 2.5 Gy**
Precancerous tissue	-	0.55 ± 0.06 Gy	0.74 ± 0.16 Gy	1.30 ± 0.17 Gy	4.7 ± 2.4 Gy	0.19 ± 0.13 Gy	**6.0 ± 2.4 Gy**
Normal tissue	-	0.55 ± 0.06 Gy	0.74 ± 0.16 Gy	1.30 ± 0.17 Gy	1.3 ± 0.4 Gy	0.19 ± 0.07 Gy	**2.6 ± 0.4 Gy**
**MID:BSA/BNCT 15 mg/kg 19 h post-injection-7.5 Gy absorbed dose to precancerous tissue**							
Tumor	-	0.67 ± 0.07 Gy	0.93 ± 0.20 Gy	1.60 ± 0.21 Gy	7.7 ± 3.1 Gy	0.24 ± 0.13 Gy	**9.3 ± 3.2 Gy**
Precancerous tissue	-	0.67 ± 0.07 Gy	0.93 ± 0.20 Gy	1.60 ± 0.21 Gy	5.9 ± 3.0 Gy	0.24 ± 0.17 Gy	**7.5 ± 3.0 Gy**
Normal tissue	-	0.67 ± 0.07 Gy	0.93 ± 0.20 Gy	1.60 ± 0.21 Gy	1.6 ± 0.5 Gy	0.24 ± 0.09 Gy	**3.2 ± 0.5 Gy**

**Table 3 life-12-01082-t003:** Percentage of tumor OR (tumor overall response = partial + complete response − CR-) after MID:BSA/BNCT (15 mg B/kg, 19 h) at each absorbed dose to precancerous tissue (Group A: 4.8 Gy, Group B: 6.0 Gy, Group C: 7.5 Gy) and control group (cancerized, not treated). n = total number of tumors.

	7 Days			14 Days			21 Days			28 Days		
	n	%OR	%CR	n	%OR	%CR	n	%OR	%CR	n	%OR	%CR
CONTROL	51	24	2	51	22	8	45	18	7	34	21	9
(A) 4.8 Gy	35	54	9	35	66	14	27	56	19	27	59	19
(B) 6.0 Gy	21	100	10	21	81	14	21	76	14	16	82	18
(C) 7.5 Gy	27	78	19	27	78	15	27	70	22	27	67	19

The hamster cheek pouch of the animals was cancerized with the optimized classical carcinogenesis protocol [36]. Once tumors developed, the animals were injected intravenously with MID:BSA (15 mg B/Kg). After 19 h, the hamster cheek pouches bearing tumors were irradiated at the RA-3 Nuclear Reactor (Buenos Aires, Argentina). The absorbed doses were prescribed to the dose-limiting tissue, the precancerous tissue surrounding the tumors. The macroscopic evaluation of the mucositis induced by BNCT and tumor volume measurements were performed at 7, 14, 21, and 28 days after BNCT. The total number of tumors evaluated underwent a reduction over time due to the occasional death of animals. Tumor response in the control group corresponds to spontaneous remissions.

**Table 4 life-12-01082-t004:** Percentage of partially responding tumors that underwent a reduction in volume by more than/equal to 50% from the value at T0 for each BNCT group. n = number of tumors with partial response (PR).

R ≥ 50%	7 Days		14 Days		21 Days		28 Days	
	n	%R	n	%R	n	%R	n	%R
(A) 4.8 Gy	16	44	18	39	10	70	11	55
(B) 6.0 Gy	19	53	14	79	13	92	11	64
(C) 7.5 Gy	16	31	17	82	13	85	13	92

The hamster cheek pouch of the animals was cancerized with the optimized classical carcinogenesis protocol [36]. Once tumors developed, the animals were injected intravenously with MID:BSA (15 mg B/Kg). After 19 h, the hamster cheek pouches bearing tumors were irradiated at the RA-3 Nuclear Reactor (Buenos Aires, Argentina). The absorbed doses were prescribed to the dose-limiting tissue, the precancerous tissue surrounding tumors. The macroscopic evaluation of the mucositis induced by BNCT and tumor volume measurements were performed at 7, 14, 21, and 28 days after BNCT.

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
