# Peer review of "Boron Neutron Capture Therapy (BNCT) Mediated by Maleimide-Functionalized Closo-Dodecaborate Albumin Conjugates (MID:BSA) for Oral Cancer: Biodistribution Studies and In Vivo BNCT in the Hamster Cheek Pouch Oral Cancer Model"

_life, 2022, doi:10.3390/life12071082_

Round 1

Reviewer 1 Report

This paper has clinical significance in the development of new drugs for BNCT. There is room for improvement in the following two points.

1)     In introduction, it is mentioned that the radiation-induced oral mucositis is a common feared side effect.  Then it is better to show the quantitative evaluation of oral mucositis after BNCT such as the marker (DOI: 10.1007/s00520-017-3783-8), if it is possible.

2)     It is necessary to consider to use “Gy” as the unit of neutron irradiation.

Author Response

We appreciate the comments and suggestions of Reviewer 1. Thank you very much for your revision. 

Related to suggestion number 1:

*In this paper our group has quantitatively assessed mucositis macroscopically, using a 5 point grading scale that our group developed based on scales used in humans and hamsters.  In this paper we quantitatively assessed severe mucositis by calculating the percentage of animals with severe mucositis (grades 4 and 5) after the dose escalation study performed with MIDBSA/BNCT. This scale allowed us to compare the mucositis induced by different BNCT protocols as was mentioned in the Discussion section (Heber et al. 2014) and was also used to demonstrate how the increasing aggressiveness of the cancerization protocol could induce a higher percentage of animals with mucositis after BNCT (Monti Hughes et al 2019). 

The quantitative method suggested by the reviewer is very interesting and undoubtedly warrants future studies in the BNCT field. However, it requires an experiment specifically designed to study this topic. Biomarkers to predict mucositis or evaluate severity of mucositis after BNCT in the hamster cheek pouch have not been studied yet. The experiment would consist of the evaluation of different time-points after BNCT: for example at 7 days, when we generally observe grade 2 mucositis, 10/14 days when we observe the peak of mucositis, and 21 days when mucositis is usually resolved. We would have to sacrifice a group of animals at each endpoint to evaluate what has been suggested. We will consider studying this topic in our future experiments, thank you very much for your suggestion. 

**It is necessary to consider to use “Gy” as the unit of neutron irradiation.

I am sorry but I did not understand this suggestion. The unit Gy has been used and mentioned each time we reported the absorbed doses prescribed to the dose-limiting tissue, the precancerous tissue. 

Please, find attached a version of the manuscript in track changes. 

Reviewer 2 Report

This is a very important article, especially in the development of the new boron drug for boron neutron capture therapy. In this study, the hamster cheek pouch oral cancer model was used to evaluate the biodistribution of MID:BSA and the therapeutic efficacy of MID:BSA/BNCT on tumors, and the associated radiotoxicity. Based on the results of the study, it was concluded that MID:BSA-mediated BNCT has a therapeutic effect in the treatment of oral cancer without obvious toxicity to the dose-limiting tissues surrounding the tumor. Moreover, the biodistribution studies also showed that the boron concentration value of precancerous tissue is higher than that of normal pocket tissue. This result would make this protocol potentially useful to achieve a therapeutic effect of tumor development inhibition. Therefore, the subject addressed in this article is worthy of investigation. Overall, this article should be accepted with minor revision for publication in Life.

There are some errors mentioned below which the review suggests the authors would revise.

1.         Regarding Table 3, please describe the brief experiment information and add it as a footnote below the Table.

2.         The content of Figure 4 has been included in Table 3, so Figure 4 can be omitted.

3.         Regarding Table 4, please describe the brief experiment information and add it as a footnote below the Table.

4.         The format of references was inconsistent, please check references carefully and revise them following the reference style of Life before resubmission.

Author Response

We appreciate the comments and suggestions of Reviewer 2. Thank you for your revision and your words related to our manuscript. 

Related to the footnotes that have to be included in tables 3 and 4, we will add this paragraph below each table: The hamster cheek pouch of the animals was cancerized with the optimized classical carcinogenesis protocol [36]. Once tumors developed, the animals were injected intravenously with MID:BSA (15 mg B/Kg). After 19h, the hamster cheek pouches bearing tumors were irradiated at the RA-3 Nuclear Reactor (BsAs, Argentina). The absorbed doses were prescribed to the dose-limiting tissue, the precancerous tissue surrounding tumors. Macroscopic evaluation of the mucositis induced by BNCT and tumor volume measurements were performed at 7, 14, 21 and 28 days after BNCT.

Figure 4 will be omitted as suggested. 

References have been checked and corrected their format. Thank you.

Please, find attached a version of the manuscript in track changes. 
